# Adult E-Cigarettes Use Associated with a Self-Reported Diagnosis of COPD

**DOI:** 10.3390/ijerph16203938

**Published:** 2019-10-16

**Authors:** Mario F. Perez, Nkiruka C. Atuegwu, Erin L. Mead, Cheryl Oncken, Eric M. Mortensen

**Affiliations:** Department of Medicine, UConn Health, University of Connecticut School of Medicine, Farmington, CT 06030, USA; atuegwu@uchc.edu (N.C.A.); mead@uchc.edu (E.L.M.); oncken@uchc.edu (C.O.);

**Keywords:** E-cigarettes, COPD, Adults, PATH

## Abstract

The use of electronic cigarettes (e-cigarettes) has increased in the US, but little is known about the effects of these products on lung health. The main purpose of this study was to examine the association between e-cigarette use and a participant’s report of being diagnosed with chronic obstructive pulmonary disease (COPD) in a nationally representative sample of adults. *Methods:* The first wave of the Population Assessment of Tobacco and Health (PATH) survey adult data was used (*N* = 32,320). Potential confounders between e-cigarette users and non-users were balanced using propensity score matching. Odds ratios (OR) were calculated to examine the association between e-cigarette use and COPD in the propensity-matched sample, the entire sample, different age groups, and in nonsmokers. Replicate weights and balanced repeated replication methods were utilized to account for the complex survey design. *Results:* Of the 3642 participants who met the criteria for e-cigarette use, 2727 were propensity matched with 2727 non e-cigarette users. In the propensity-matched sample, e-cigarette users were more likely to report being diagnosed with COPD (OR 1.43, 95% confidence interval [CI] 1.12–1.85) than non-e-cigarette users after adjusting for confounders. The result was similar in the entire sample and in the different age subgroups. Among nonsmokers, the odds of reporting a COPD diagnosis were even greater among e-cigarette users (OR 2.94, 95% CI 1.73–4.99) compared to non-e-cigarette users. *Conclusion:* Our findings demonstrate that e-cigarette use was associated with a reported diagnosis of COPD among adults in the US. Further research is necessary to characterize the nature of this association and on the long-term effects of using e-cigarettes.

## 1. Introduction

Electronic cigarette (e-cigarette) use is increasing exponentially in the United States, especially among adolescents and young adults [1,2]. Approximately 20% of adolescents and 8% of adults between the age of 18 and 24 years old currently use e-cigarettes in the US [2,3]. E-cigarettes are devices powered by a battery (usually made out of lithium) that heats up a metallic coil embracing a wick and aerosolizes a solution commonly known as “e-juice” or “e-liquid” for the user to inhale. The e-juice usually contains propylene glycol, vegetable glycerin, flavors and nicotine or a mixture of these chemicals [4]. The exact composition of the “e-juice” varies by manufacturer and is often challenging to discern [5]. Although e-cigarettes have been generally perceived as safer than conventional cigarettes in terms of various toxicant exposures [6,7], a need exists to examine the potential risks of e-cigarettes per se on various aspects of human health and disease, especially lung health, as e-cigarette aerosol has direct contact with the lungs and there are growing reports of acute and subacute pulmonary illnesses associated with their use [8].

Smokers with lung disease may consider e-cigarettes to be a tool to help them quit smoking or believe that switching from conventional cigarettes to e-cigarettes improves their respiratory symptoms [9,10,11,12]. Although evidence of e-cigarettes as a smoking cessation tool continues to grow [13,14,15], the safety profile of e-cigarettes is being currently challenged by health authorities in the US as evidence of their negative effects is starting to appear [8,16]. Furthermore, many in the medical community, including physicians in the field of pulmonary medicine, have had concerns that e-cigarettes may be detrimental to lung health among individuals who have never smoked conventional cigarettes and/or who are former smokers [17,18]. These concerns are based on prior studies showing an association between e-cigarette use and asthma [19,20,21,22] and more recently, reports linking the use of e-cigarettes with respiratory failure and death [8].

Chronic obstructive pulmonary disease (COPD) is one of the leading causes of morbidity and mortality both in the US and globally [23,24]. Smoking conventional cigarettes and exposure to second-hand tobacco smoke are the main known risk factors for developing COPD [25,26]; to our knowledge, it remains to be determined whether e-cigarette use is associated with COPD diagnosis. In-vitro and animal studies have shown that exposure to aerosols from e-cigarettes (referred as “vapors”) triggers an inflammatory response similar to that seen with conventional cigarette smoke [27,28,29,30], which has been linked with the development of chronic respiratory illnesses, such as asthma and COPD [31].

A recent epidemiological study on the islands of Hawaii reported an association between the use of e-cigarettes, asthma and COPD among adults [32]. However, this could be the result of intrinsic characteristics of this population, since other longitudinal reports suggest that the use of e-cigarettes do not lead to significant changes in pulmonary function tests, fraction of exhaled nitric oxide (a measurement of airway inflammation) or lung imaging findings suggestive of early COPD [33].

The main purpose of this study was to determine whether an association exists between the use of e-cigarettes and reporting a diagnosis of COPD in a representative sample of adults [10]. We hypothesize that individuals who use e-cigarettes have greater odds of reporting a COPD diagnosis.

## 2. Materials and Methods

For this study, publicly available adult data was used from the first wave of the Population Assessment of Tobacco and Health (PATH) cohort study conducted from 12 September 2013 to 15 December 2014 [34]. This nationally representative study was conducted with civilian non-institutionalized adults (ages ≥18 years) and youths (ages 12–17 years) in the US. The survey used both computer-assisted personal interviewing and audio computer-assisted self-interviewing to collect information such as demographics, tobacco use patterns, tobacco initiation and health outcomes. Surveys from 32,320 adults were used for this analysis. Weighting procedures adjusting for varying selection probabilities and differential non-response rates were included in the study design. Further details regarding the PATH study design and methods are published elsewhere [34,35].

### 2.1. Measures of Independent Variables

For this analysis, the following potential confounders were examined: respondents’ demographic characteristics, measures of health, secondhand smoke (SHS) exposure, years of conventional cigarette use, current and former conventional tobacco use status, other tobacco product use, and history of exposure to heroin, inhalants or hallucinogens. Demographic characteristics of interest included age (18–24, 25–34, 35–44, 45–54, 55–64 and 65 years or older), sex, race (white, black, and other), ethnicity (Hispanic and not Hispanic), poverty level (100%, 100–199%, and 200% of poverty guideline), census region (Northeast, Midwest, South, and West) and highest educational grade achieved (less than high school [HS], general education diploma [GED], HS graduate, some college/associates degree, and Bachelor’s degree or higher). Other tobacco product use was defined as ever using traditional or filtered cigars, cigarillos, pipe, hookah, oral tobacco, and cigars with marijuana (blunts). Because of their associations with COPD, the following measures of respondents’ health were examined: body mass index (BMI) and self-reported histories of asthma, high blood pressure, high cholesterol, congestive heart failure, stroke, heart attack, and diabetes. Childhood SHS exposure and current SHS exposure in the home were also included. Current conventional cigarette smoker was defined as having smoked more than 100 cigarettes in lifetime, and currently smoking every day or some days. Former conventional cigarette smoker was defined as having smoked more than 100 cigarettes in lifetime, and currently not smoking. Participants who have ever smoked a cigarette “even one or two puffs” were asked for their lifetime number of cigarettes smoked. Nonsmokers were defined as those who smoked fewer than 100 cigarettes in their lifetime, a standard definition used by the Centers for Disease Control and Prevention [36]. The variable “years of cigarette use” was not collected for subjects who were nonsmokers or reported never smoking cigarettes fairly regularly and as such, those subjects were assigned a value of 0 for years of conventional cigarette use. E-cigarette use was determined as follows: respondents were first asked “if they had ever used an e-cigarette, even one or two times”. Those who responded “yes” were subsequently asked “if they now used e-cigarettes every day, some days, or not at all”. E-cigarette users were defined as those who reported ever using e-cigarettes and currently using e-cigarettes every day or some days. All the other respondents who did not meet this criterion were considered “non-users of e-cigarettes” for the purpose of this study (Figure 1). This is the definition of e-cigarette user in the PATH survey data and the same definition has been used for the analysis PATH survey data in a scientific publication by Coleman et al. [10].

### 2.2. Measure of Dependent Variable

Individuals who reported COPD were defined as those who answered yes to any of the following three questions: “Has a doctor, nurse or other health professional ever said you had (1) COPD? (2) Emphysema? Or (3) Chronic bronchitis?”

### 2.3. Statistical Analysis

For our primary analyses, propensity score matching was used to balance confounders between groups (e-cigarette users and non-e-cigarette users). Propensity score matching helped reduce bias by identifying controls (non-e-cigarette users) who are matched in probability on a large number of potential confounders to cases (e-cigarette users), thereby increasing between-group comparability [37]. Propensity score matching for survey data was performed as described by Lenis et al [38]. Logistic regression was used to create the propensity score and then optimal matching with no replacement was performed. The variables included in the propensity score are displayed in Table 1.

Odds ratios (ORs) were calculated on the propensity matched data to determine the association between e-cigarette use and reporting COPD using logistic regression models. In addition, logistic regression models were used to determine the association between e-cigarette use and reporting COPD in the following groups: the entire cohort, subjects 35 years and older, subjects 45 years and older, subjects 55 years and older and nonsmokers. For the analysis with nonsmokers, the lifetime number of cigarettes smoked (0 to <100 cigarettes) instead of years of cigarette use was used as a covariate in the model. All analyses were conducted using replicate weights and balanced repeated replication methods to account for PATH’s survey design. Respondents that had missing data were removed from the analysis. All analyses were conducted with R version 3.4.2 [39]. Statistical significance was set as a two-tailed *p*-value < 0.05.

## 3. Results

### 3.1. Study Sample Characteristics

In the study population (*N* = 32,320), 3642 respondents were current e-cigarette users and 28,606 were non e-cigarette-users (hereinafter referred as “non-users”). Demographic characteristics and tobacco product use patterns for this population have been previously reported by Coleman et al. [10]. Figure 1 depicts the number of respondents included in the analysis, distributed according to current e-cigarette use. Among the e-cigarette users, 734 were daily e-cigarette users and 2908 reported using e-cigarettes some days, 69.7% (95% confidence interval [CI] 68.0–71.4) were current smokers, 14.3% (95% CI 12.9–15.7) were former smokers and 16.0% (95% CI 14.9–17.1) were nonsmokers. Among daily e-cigarette users, 49.6% (95% CI 45.7–53.5) were current smokers, 41.8% (95% CI 37.9–45.8) were former smokers and 8.5% (95% CI 6.4–10.6) were nonsmokers, and among someday users, 75.1% (95% CI 73.4–76.8) were current smokers, 6.8% (95% CI 5.8–7.9) were former smokers and 18.0% (95% CI 16.6–19.4) were nonsmokers.

Among e-cigarette users, 20.9% (95% CI 19.5–22.3) were 18 to 24 years old, 26.5% (95% CI 24.8–28.1) were 25 to 34 years old, 18.8% (95% CI 17.3–20.3) were 35 to 44 years old, 16.6% (95% CI 15.3–17.9) were 45 to 54 years old, and 12.4% (95% CI 11.2–13.6) were 55-64 years old and 4.9% (95% CI 3.9–5.8) were 65 years and older.

The prevalence of current e-cigarette use was 5.5% (95% CI 5.3–5.8) and the prevalence of reporting COPD was 5.9% (95% CI 5.5–6.3) for the entire population. The prevalence of reporting COPD among the nonsmokers was 3.0% (95% CI 2.7–3.4). The prevalence of reporting COPD among e-cigarette users who were also current users of conventional cigarettes was 11.5% (95% CI 9.8–13.1). Figure 2 shows the prevalence of reporting COPD for current e-cigarette users and non-e-cigarette users according to their cigarette smoking status.

A total of 15,854 participants met the definition of nonsmokers and among those, 44.1% (95% CI 43.6–44.7) were men. A total of 16,327 respondents were considered smokers. In this group, 54.3% (95% CI 53.4–55.2) were men and the average length of use of conventional cigarettes was 17.9 years (95% CI 17.5–18.3). The prevalence of reporting COPD among conventional cigarette smokers was 10.5% (95% CI 9.8–11.2). Among e-cigarettes users, 53.5% (95% CI 51.5–55.4) were men, 11.1% (95% CI 9.7–12.5) reported COPD and the average length of use of conventional cigarettes was 16.1 years (95% CI 15.5–16.6).

### 3.2. Association between E-Cigarette Use and Reporting COPD

After propensity matching, there were 2727 e-cigarette users and 2727 non-users. The balance between the covariates are presented in Table 1. There were no statistically significant differences in the key covariates between the two groups in either the sample or in the entire population.

In the propensity matched analysis, e-cigarette users had greater odds of reporting COPD than non-users (OR 1.43, 95%CI 1.12–1.85). The use of e-cigarettes was also positively associated with reporting COPD (OR 1.47, 95%CI 1.21-1.79) in the entire cohort and nonsmokers (OR 2.94, 95%CI 1.73–4.99) (Table 2). The use of e-cigarettes was also positively associated with reporting a diagnosis of COPD in subjects that were 35 years and older (OR 1.57, 95%CI 1.27–1.95), subjects that were 45 years and older (OR 1.57, 95%CI 1.20–2.06) and subjects 55 years and older (OR 1.62, 95%CI 1.14–2.31). Among nonsmokers of cigarettes, e-cigarette users had almost three-fold greater odds of reporting COPD than non-users (OR 2.94, 95% CI 1.73–4.99). In the entire cohort, daily e-cigarette users (OR 1.59, 95%CI 1.06–2.37), someday e-cigarette users (OR 1.97, 95%CI 1.55–2.49) and former e-cigarette users (OR 1.73, 95%CI 1.46–2.06) all had increased odds of reporting a diagnosis of COPD compared to participants who reported never having used an e-cigarette.

## 4. Discussion

We found a significant association between using e-cigarettes every day or somedays and the reported diagnosis of COPD, even after adjusting for the use of combustible tobacco products and other risk factors associated with this condition. The subgroup analysis showed an association between the use of e-cigarettes and reporting COPD among subjects 35 years and older, 45 years and older and 55 years and older. There were also increased odds of a reporting COPD for a subgroup of respondents who used e-cigarettes and not combustible cigarette smokers.

Our study adds to the literature by examining the association between e-cigarette use and reported COPD in a large population that is representative of the adults in the US. This association is supported by studies that have shown a decline in lung function with exposure to aerosol from e-cigarettes and similarities in the biological response to conventional tobacco cigarette smoke [27,28,29,40]. In humans, at least one study has reported an association between e-cigarette use and symptoms of chronic bronchitis [41], and another study using the COPDGene and SPIROMICS cohorts found an association between e-cigarette use and progression of COPD among subjects at risk or in whom the disease had already been established [42]. Additionally, recent work by Wills et al. [32] showed a similar association between e-cigarette use and respiratory diseases among adults in the state of Hawaii. More recently, a subacute pulmonary illness has been described among users of e-cigarettes [8]; however, the etiology of this acute illness is still unclear and undergoing scientific investigation [16,43].

It is noteworthy that we are not suggesting that e-cigarettes may not be beneficial for smoking cessation, but rather that they may pose a significant risk for lung health compared to not using e-cigarettes. An internet-based survey of asthmatics and persons diagnosed with COPD showed improvements in pulmonary symptoms in formerly smoker e-cigarette users compared with cigarette smokers [44]. Another study found subjective and objective improvements in respiratory outcomes in patients with COPD who reduced their tobacco consumption after switching to E-cigarettes [45,46]. Despite their potential as a smoking cessation tool, our findings add to the literature by showing that e-cigarettes may pose an increased risk of chronic pulmonary disease compared to not using e-cigarettes.

Our study had several strengths and limitations. An important strength was the large number of respondents available for analysis, which allowed us to compare the relationship between the use of e-cigarettes and the reported diagnosis of COPD after controlling for the effects of using combustible tobacco products and other risks factors. In addition, unlike the data presented by Wills et al. [32], data from the PATH survey is considered representative of the entire US and is consistent with similar national surveys [47]. These strengths increase the generalizability of our findings to US adults.

A limitation was the cross-sectional nature of the analysis, which did not allow us to establish a causal relationship between e-cigarette use and reporting COPD. Although we controlled for potential confounders, it is possible that there were differences between groups that were not accounted for in the analyses. One of the confounders adjusted for was smoking duration as pack-years was not readily available in the publicly available data; however, smoking duration may be a better predictor of obstructive pulmonary disease [48]. Moreover, we do not have information on the number of participants that stopped e-cigarette and/or cigarette use or switched from cigarettes to e-cigarettes and vice versa after a diagnosis of COPD. Another significant limitation is the lack of pulmonary function tests and/or spirometry necessary to objectively make the diagnosis of COPD; this is a common barrier for large surveys, including those used to determine the national prevalence of COPD. Furthermore, the diagnosis of COPD and related conditions, such a chronic bronchitis and emphysema, could be affected by recall bias. However, the prevalence of disease in this population (5.9%) is similar to the general prevalence reported in the US [24,49].

Our work provides evidence that the use of e-cigarettes is associated with having a reported diagnosis of COPD. More research is necessary to assess the long-term effects and safety of these products. Information is urgently needed as government agencies attempt to regulate the e-cigarette manufacturing industry, and clinicians who are on the front lines caring for cigarette smokers and counseling them about smoking cessation options, need more information to properly advise their patients.

## 5. Conclusions

Our findings demonstrate that e-cigarette use is associated with a reported diagnosis of COPD among adults in the US. Further research is necessary to characterize the nature of this association and both the respiratory and systemic long-term effects of using e-cigarettes.

## Figures and Tables

**Figure 1 ijerph-16-03938-f001:**
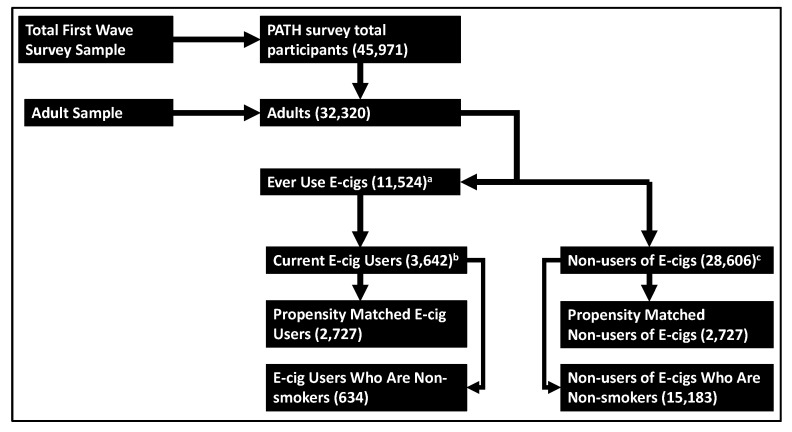
Selection of population sample for analysis. ^a^ Respondents who affirmatively answered the question: “Have you used an e-cigarette, such as NJOY, Blu, or Smoking Everywhere, even one or two times?” ^b^ Respondents who responded “every day” or “some days” to: “Do you now use e-cigarettes?” ^c^ Includes respondents who: (1) responded “no” to “Have you used an e-cigarette, such a NJOY, Blu, or Smoking Everywhere, even one or two times?”, or (2) responded “not at all” to “Do you now use e-cigarette?”.

**Figure 2 ijerph-16-03938-f002:**
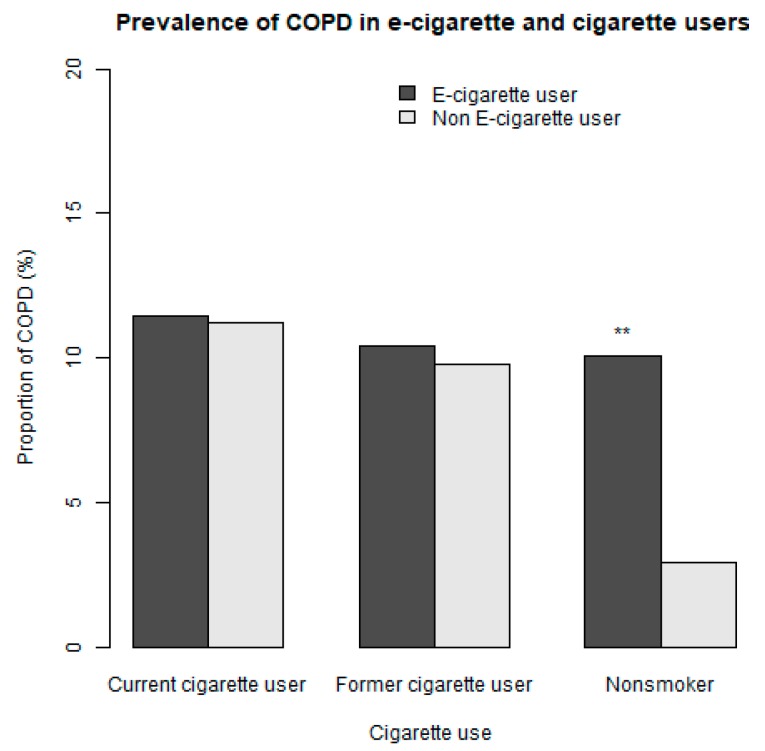
Prevalence of reporting being diagnosed with chronic obstructive pulmonary disease (COPD) for current e-cigarette users and non-e-cigarette users that are current, former and nonsmokers of cigarette. There was no statistical difference in the prevalence of COPD for e-cigarette users and non-users that were current (*p* = 0.8) and former cigarette users (*p* = 0.7). There was a statistically significant difference in the prevalence of COPD for e-cigarette users and non-users that were nonsmokers (*p* < 0.001). ** Statistically significant.

**Table 1 ijerph-16-03938-t001:** Demographics and health characteristics of propensity matched population.

Variable	Controls ^a^	E-Cigarette Users ^a^	Sample Balance		Population Balance ^c^	
			*p*-Value	SMD ^b^	*p*-Value	SMD ^b^
Number of Individuals	2727	2727				
BMI	27.5 (6.7)	27.7 (6.8)	0.38	0.02	0.74	0.01
Years of conventional cigarette use	14.2 (14.4)	13.9 (13.7)	0.37	0.02	0.80	0.01
**Age Group**			0.85	0.04	0.86	0.04
18 to 24 years old	793 (29.1)	830 (30.4)				
25 to 34 years old	685 (25.1)	685 (25.1)				
35 to 44 years old	475 (17.4)	478 (17.5)				
45 to 54 years old	419 (15.4)	394 (14.4)				
55 to 65 years old	265 (9.7)	256 (9.4)				
65 years old or older	90 (3.3)	84 (3.1)				
Male	1493 (54.7)	1451 (53.2)	0.27	0.03	0.15	0.04
Hispanic	396 (14.5)	396 (14.5)	1.00	<0.001	0.92	0.003
**Race**			0.78	0.02	0.12	0.06
White	2171 (79.6)	2155 (79.0)				
Black	257 (9.4)	272 (10.0)				
Other	299 (11.0)	300 (11.0)				
**Grade of Highest Education**			0.37	0.06	0.29	0.07
Less than High School	312 (11.4)	306 (11.2)				
GED	289 (10.6)	271 (9.9)				
High school graduate	633 (23.2)	664 (24.3)				
Some college (no degree) or associates degree	1096 (40.2)	1131 (41.5)				
Bachelor’s degree or higher	397 (14.6)	355 (13.0)				
**Census Region**			0.84	0.03	0.75	0.04
Northeast	301 (11.0)	310 (11.4)				
Midwest	718 (26.3)	692 (25.4)				
South	1059 (38.8)	1081 (39.6)				
West	649 (23.8)	644 (23.6)				
Second Hand Exposure During Childhood	1747 (64.1)	1767 (64.8)	0.59	0.02	0.72	0.01
History of Asthma	356 (13.1)	371 (13.6)	0.58	0.02	0.78	0.01
**Poverty level**			0.69	0.02	0.48	0.04
100% of poverty guideline	985 (36.1)	987 (36.2)				
100-199% of poverty guideline	674 (24.7)	698 (25.6)				
≥200% of poverty guideline	1068 (39.2)	1042 (38.2)				
History of Blunt Use	1097 (40.2)	1107 (40.6)	0.80	0.01	0.29	0.03
Current Secondhand Smoke Exposure in Household	1366 (50.1)	1351 (49.5)	0.71	0.01	0.82	0.01
Never Used Cigars ^d^	1193 (43.7)	1197 (43.9)	0.94	0.003	0.40	0.03
Never Used Cigarillos	968 (35.5)	969 (35.5)	1.00	0.001	0.97	0.001
Never Used Pipe	1762 (64.6)	1773 (65.0)	0.78	0.01	0.40	0.03
Never Used Hookah	1442 (52.9)	1412 (51.8)	0.43	0.02	0.48	0.03
Never Used Oral Tobacco	1729 (63.4)	1703 (62.4)	0.48	0.02	0.81	0.01
Current conventional cigarette use	1880 (68.9)	1899 (69.6)	0.60	0.02	0.08	0.05
Former conventional cigarette use	395 (14.5)	368 (13.5)	0.31	0.03	0.11	0.05
History of High Blood Pressure	519 (19.0)	527 (19.3)	0.81	0.01	0.32	0.03
History of High Cholesterol	376 (13.8)	369 (13.5)	0.81	0.01	0.98	0.001
History of Congestive Heart Failure	35 (1.3)	42 (1.5)	0.75	0.02	0.67	0.01
History of Stroke	47 (1.7)	43 (1.6)	0.52	0.01	0.92	0.003
History of Heart Attack	40 (1.5)	47 (1.7)	0.75	0.02	0.85	0.01
History of Heroin, Inhalants or Hallucinogens	482 (17.7)	462 (16.9)	0.50	0.02	0.58	0.02
History of Diabetes	257 (9.4)	261 (9.6)	0.89	0.01	0.98	0.001

^a^ Reported as frequency values (or proportions) or mean (and standard deviation) as appropriate. These reflect unweighted data. ^b^ Standardized mean difference; ^c^ Population balance reflect weighted *p* values and SMD. ^d^ Respondents have not used either traditional or filtered cigars.

**Table 2 ijerph-16-03938-t002:** Results from the multiple logistic regression models showing the odds ratio (95% confidence intervals) of reported chronic obstructive pulmonary disease (COPD) for e-cigarette users vs. non-users for different group categories and by age.

Category	Entire Cohort	Nonsmokers ^a^	Subjects ≥35 ^b^	Subjects ≥45 ^b^	Subjects ≥55 ^b^
**Not users**	**Reference**	**Reference**	**Reference**	**Reference**	**Reference**
**E-cigarette user**	1.47 (1.21–1.79)	2.94 (1.73–4.99)	1.57(1.27–1.95)	1.57 (1.20–2.06)	1.62 (1.14–2.31)
**Age groups ^b^**					
**18 to 24**	Reference	Reference			
**25 to 34**	1.03 (0.70–1.52)	1.29 (0.68–2.42)			
**35 to 44**	1.79 (1.21–2.63)	1.97 (0.87–4.45)	Reference	-	-
**45 to 54**	2.37(1.67–3.37)	1.96 (1.03–3.71)	1.33 (0.96–1.85)	Reference	-
**55 to 65**	3.15 (2.18–4.54)	4.06 (2.07–7.97)	1.79 (1.25–2.58)	1.35 (1.03–1.77)	Reference
**>65**	3.79 (2.55–5.65)	5.96 (3.31–10.74)	2.15 (1.38–3.37)	1.64 (1.18–2.26)	1.25 (0.91–1.72)

^a^ All confounders listed in Table 1 were included in the regression model except for years of conventional cigarettes use, current and former conventional cigarette use. ^b^ Years old.

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
