# Peer review of "Adult E-Cigarettes Use Associated with a Self-Reported Diagnosis of COPD"

_ijerph, 2019, doi:10.3390/ijerph16203938_

Round 1
Reviewer 1 Report
In this article the authors examined the relationship between usage of electronic cigarettes with self-reported chronic obstructive pulmonary disease (COPD) and found these two to be positively correlated. It is clear that the authors were aware of the difference between association and causation due to the cross-sectional design of the data they have used. Therefore, the article title, as well as its opposite statement like “Adults with a Diagnosis of COPD are More Likely to Use E-Cigarettes” are both misleading and should be revised.
The paragraph in lines 141-150 made me confused. The distinction between user and non-users and between smoker and nonsmoker was not clear to me. What did it mean “Of the non-users…734 of the respondents were daily e-cigarette users and 2908 reported 143 using some days”? How could non-users of electronic cigarettes still be daily e-cigarette users? Maybe a tree diagram could depict the composition more clearly.
The authors used “e-cig” and “e-cigarette” simultaneously. One of them is good enough.
Author Response
We want to thank reviewer number one for the valuable input provided in order to make our manuscript better. We addressed the insightful comments as follows:
In this article the authors examined the relationship between usage of electronic cigarettes with self-reported chronic obstructive pulmonary disease (COPD) and found these two to be positively correlated. It is clear that the authors were aware of the difference between association and causation due to the cross-sectional design of the data they have used. Therefore, the article title, as well as its opposite statement like “Adults with a Diagnosis of COPD are More Likely to Use E-Cigarettes” are both misleading and should be revised.
We have modified the title to “Adult E-Cigarettes Use Associated with a Self-Reported Diagnosis of COPD”.
The paragraph in lines 141-150 made me confused. The distinction between user and non-users and between smoker and nonsmoker was not clear to me. What did it mean “Of the non-users…734 of the respondents were daily e-cigarette users and 2908 reported 143 using some days”? How could non-users of electronic cigarettes still be daily e-cigarette users? Maybe a tree diagram could depict the composition more clearly.
We apologize for the confusion and we have edited the paragraph for clarity.
“In the study population (N=32,320), 3,642 respondents were current e-cigarette users and 28,606 were non e-cigarette-users (hereinafter referred as “non-users”). Demographic characteristics and tobacco product use patterns for this population have been previously reported by Coleman et al.[10] Figure 1 depicts the number of respondents included in the analysis distributed according to current e-cigarette use. Among the e-cigarette users, 734 were daily e-cigarette users and 2908 reported using e-cigarettes some days, 69.7% (95% confidence interval [CI] 68.0-71.4) were current smokers, 14.3% ( 95% CI 12.9 - 15.7) were former smokers and 16.0 (95% CI 14.9-17.1) were nonsmokers. Among daily e-cigarette users 49.6% (95% CI 45.7- 53.5) were current smokers, 41.8% (95% CI 37.9-45.8) were former smokers and 8.5% (95% CI 6.4-10.6) were nonsmokers, and among someday users 75.1% (95% CI 73.4-76.8) were current smokers, 6.8% (95% CI 5.8 -7.9) were former smokers and 18.0% (95% CI 16.6-19.4) were nonsmokers.”
The authors used “e-cig” and “e-cigarette” simultaneously. One of them is good enough.
We agree with the reviewer and appreciate the keen eye and close attention. We favor using e-cigarette and therefore will change all “e-cig” references to the preferred “e-cigarette” abbreviation.
Reviewer 2 Report
This paper presents results from an analysis of the 2013-2014 PATH study in the US which included 32,320 respondents ages 18+. Three types of analyses were conducted regarding reports of ever having a diagnosis of COPD: a propensity matched sample of current e-cigarette users compared to non-users (N’s=2727), tests between e-cigarette users and users of regular cigarettes, and logistic regressions predicting COPD with current use of e-cigarettes as the main predictor. The propensity analysis indicated that current use of e-cigarettes was a predictor of COPD (OR = 1.43). A comparison between users of e-cigarettes and those who did not use either e-cigarettes or regular cigarettes also showed increased levels of COPD. However, use of e-cigarettes among former regular cigarette users revealed no increased level of COPD attributable to current use of e-cigarettes (Figure 2). Finally, the regressions using any use of e-cigarettes indicated strong associations between e-cigarette use and COPD (Table 2).
The authors conclude that there is an association between e-cigarette use and COPD that requires further study. The analyses that most clearly support this conclusion are the propensity modeled data and the comparison with non-users of regular cigarettes, who presumably have limited exposure to tobacco smoke other than the vaping experience. The finding that current e-cigarette use among former regular cigarette users was not associated with increased COPD suggests that the use of e-cigarettes does not increase the risk of COPD for those who may have used regular cigarettes in the past. Thus, if e-cigarettes lead to a reduction in regular smoking, there may be a net benefit for those users. However, for those who are using e-cigarettes without any prior use of regular cigarettes, the risk may be more than doubled. The authors should comment on this possibility. They should also report the odds ratio for this finding in the results as they do in the abstract.
Regarding the regressions reported in Table 2, I find these analyses less than convincing. First, they appear to regress COPD on any us of e-cigarettes, when it would be desirable to show a dose-response relation using three levels of current use: every day, some days, or not at all. It might even make sense to include a fourth level of having ever used e-cigarettes in the past. Second, it is not clear why current use of regular cigarettes was not included in the models when it was in the other analyses.
There is also a lack of clarity regarding how lifetime number of regular cigarettes used was obtained for the analysis of e-cigarette use among non-smokers (lines 129-130).
The finding mentioned in the Discussion (lines 214-216) also seems to relate to the question of whether use of e-cigarettes may help those who quit the use of regular cigarettes. In other words, even if e-cigarettes are harmful to those who never smoked, they may have some benefits for those who have managed to quit smoking.
There are some places where the text needs attention. Line 147: 16.0 %. Line 153: was and 5.9%. lines 156-157: “dual users” are never referenced again. Line 180: that result is not in Table 2. Line 181: CI (.21, 1.79).
Author Response
We want to thank reviewer number two for the valuable comments and input provided in order to make our manuscript better. We have addressed these insightful comments below:
This paper presents results from an analysis of the 2013-2014 PATH study in the US which included 32,320 respondents ages 18+. Three types of analyses were conducted regarding reports of ever having a diagnosis of COPD: a propensity matched sample of current e-cigarette users compared to non-users (N’s=2727), tests between e-cigarette users and users of regular cigarettes, and logistic regressions predicting COPD with current use of e-cigarettes as the main predictor. The propensity analysis indicated that current use of e-cigarettes was a predictor of COPD (OR = 1.43). A comparison between users of e-cigarettes and those who did not use either e-cigarettes or regular cigarettes also showed increased levels of COPD. However, use of e-cigarettes among former regular cigarette users revealed no increased level of COPD attributable to current use of e-cigarettes (Figure 2). Finally, the regressions using any use of e-cigarettes indicated strong associations between e-cigarette use and COPD (Table 2).
We appreciate the input from reviewer 2, but would like to point out that Figure 2 only presents the prevalence of reporting the diagnosis of COPD among the different subpopulations. It does not show the results of a regression model comparing the e-cigarette use in those groups. The authors conclude that there is an association between e-cigarette use and COPD that requires further study. The analyses that most clearly support this conclusion are the propensity modeled data and the comparison with non-users of regular cigarettes, who presumably have limited exposure to tobacco smoke other than the vaping experience. The finding that current e-cigarette use among former regular cigarette users was not associated with increased COPD suggests that the use of e-cigarettes does not increase the risk of COPD for those who may have used regular cigarettes in the past. Thus, if e-cigarettes lead to a reduction in regular smoking, there may be a net benefit for those users. However, for those who are using e-cigarettes without any prior use of regular cigarettes, the risk may be more than doubled. The authors should comment on this possibility. They should also report the odds ratio for this finding in the results as they do in the abstract.
We do agree that further research is needed in regards to the association we report between the use of e-cigarettes and reporting ever being told that you have COPD by a health care provider. Similarly to the fact that we can’t infer a casual association between e-cigarette use and COPD, we have to also be cautious on inferring that former cigarette users that currently use e-cigarettes have no increased risk of developing COPD from e-cigarettes. Some of the former cigarette users may have been dual users of e-cigarettes and cigarettes before the diagnosis of COPD.
The odds ratio for association between COPD and e-cigarettes for non-smokers is in table 2 but we have included this as text in the results section in the manuscript.
Regarding the regressions reported in Table 2, I find these analyses less than convincing. First, they appear to regress COPD on any use of e-cigarettes, when it would be desirable to show a dose-response relation using three levels of current use: every day, some days, or not at all. It might even make sense to include a fourth level of having ever used e-cigarettes in the past. Second, it is not clear why current use of regular cigarettes was not included in the models when it was in the other analyses.
We apologize for the confusion. Current use of cigarettes was included in all models except for the multivariable model for nonsmokers of cigarettes (who are not current cigarette users). The table has a subscript “ a ” denoting this for only nonsmokers.
We thank the reviewer for this suggestion. We calculated the odds ratio of COPD based on self-reported frequency of e-cig use. In the entire cohort, daily e-cigarette users (OR 1.59, 95%CI 1.06-2.37), someday e-cigarette users (OR 1.97, 95%CI 1.55-2.49) and former e-cigarette users (OR 1.73, 95%CI 1.46-2.06) all had an increased odds of COPD compared to participants who reported never having used an e-cigarette. We have included this in the manuscript.
There is also a lack of clarity regarding how lifetime number of regular cigarettes used was obtained for the analysis of e-cigarette use among non-smokers (lines 129-130).
We apologize for the confusion. We have included this statement in the method section. “Participants who have ever smoked a cigarette “even one or two puffs” were asked for lifetime number of cigarettes smoked.” We have also changed the statement in the statistical analysis section to “For the analysis with nonsmokers, lifetime number of cigarettes smoked (0 to <100 cigarettes) instead of years of cigarette use was used as a covariate in the model.”
The finding mentioned in the Discussion (lines 214-216) also seems to relate to the question of whether use of e-cigarettes may help those who quit the use of regular cigarettes. In other words, even if e-cigarettes are harmful to those who never smoked, they may have some benefits for those who have managed to quit smoking.
We agree with the reviewer that there is literature supporting the use of e-cigarettes as an alternative to conventional cigarette smoking, but one has to be cautious as the long-term consequences of using these products are still unclear. Furthermore, recent developments, with deaths associated to lung injury caused by e-cigarettes among never smokers, former smokers and current smokers of conventional cigarettes call raise significant concerns about e-cigarettes.
There are some places where the text needs attention. Line 147: 16.0 %. Line 153: was and 5.9%. lines 156-157: “dual users” are never referenced again. Line 180: that result is not in Table 2. Line 181: CI (.21, 1.79).
We thank the reviewer for their attention to details. In line 147 the “%” signed has been entered next to the number 16, so it will read: “… and 16.0% (95% CI 14.9-17.1) were nonsmokers”. In line 153 the word “and” has been removed. In line 156-157 the parenthesis alluding to “dual users” has been removed. In line 181 a “1” has been entered preceding .21, reflecting the values presented in Table 2.
Reviewer 3 Report
Dear authors,
It is a well written, interesting manuscript that will add to the existing Knowledge on e-cigarette health effects. In overall I would like to stress the conceptual difference between the majority of the scientific community and the British practice regarding e- cigarette use and state that e- cigarette is not a cessation tool, it is a tobacco product and it is addictive through it's nicotine: (http://apps.who.int/gb/fctc/PDF/cop6/FCTC_COP6_10Rev1-en.pdf?ua=1), (https://www.chestnet.org/News/Press-Releases/2018/05/tob).
That said, I believe that the manuscript should undergo some minor changes, especially in the introduction and discussion sections, in order to comply with this concept. My comments are listed below:
L. 38 : perceived as , instead of considered
L. 45: growing evidence of negative health effects (relevant literature referred)
L. 48 concern/ evidence of adverse health effects instead.
L. 56: vapor/ it is an aerosol and termed as such
L. 64: The main purpose of this study was
L.184: it is more accurate to read : in comparison to non smokers, e-cig users... (e-cigarette users are not by any means non smokers)
L.197 : it is suggested to read:
for respondents who were e-cigarette users and not combustible cigarette smokers
L.200: add clinical studies on lung function alterations by e-cigarette use (Respirology. 2018;23(3):291–297. doi: 10.1111/resp.13180)
L.208-209: suggested to read: however the etiology of this acute illness is still unclear and under ongoing scientific investigation
L.210: it is not a cessation tool, since it promotes addiction therefore it is suggested to read as: In addition to previous studies, the present study shows that e-cigarettes pose a significant risk for lung health 
(relevant refference needed)
L.212 -214: it is suggested not to compare e-cigarette with combustible cigarette, but to refer instead clinical studies that show e-cigarette effects on lung function such as:Respirology. 2018;23(3):291–297. doi: 10.1111/resp.13180, Toxicol. Appl. Pharmacol. 2014; 278: 9– 15. and Tob. Prev. Cessat. 2017; 3: 1-8
Author Response
We want to thank reviewer number 3 for providing suggestion and comments to greatly improve the quality of our manuscript. We have addressed the comments and suggestions as follows:
Dear authors,
It is a well written, interesting manuscript that will add to the existing Knowledge on e-cigarette health effects. In overall I would like to stress the conceptual difference between the majority of the scientific community and the British practice regarding e- cigarette use and state that e- cigarette is not a cessation tool, it is a tobacco product and it is addictive through it's nicotine: (http://apps.who.int/gb/fctc/PDF/cop6/FCTC_COP6_10Rev1-en.pdf?ua=1), (https://www.chestnet.org/News/Press-Releases/2018/05/tob).
That said, I believe that the manuscript should undergo some minor changes, especially in the introduction and discussion sections, in order to comply with this concept. My comments are listed below:
38 : perceived as , instead of considered
We agree with this suggestion and “perceived as” will be used, instead of considered.
45: growing evidence of negative health effects (relevant literature referred)
That sentence will be modified as: “the safety profile of e-cigarettes is currently being challenged by health authorities in the US as evidence of their negative effects is starting to appear.”
48 concern/ evidence of adverse health effects instead.
The reference cited makes allusion to surveys performed before the reports of VAPI.
56: vapor/ it is an aerosol and termed as such
We agree with the reviewer and allusions to vapor will be changed to aerosol for clarity and better representation of the material produced by e-cigarettes.
64: The main purpose of this study was
The verb to be has been changed from the present to the past tense “was”.
L.184: it is more accurate to read : in comparison to non smokers, e-cig users... (e-cigarette users are not by any means non smokers)
We apologize for the confusion. We are reporting the results for e-cigarette use in nonsmokers of cigarettes so we are not comparing to nonsmokers. For clarity, we have changed the text to include “nonsmokers of cigarettes”L.197 : it is suggested to read:
for respondents who were e-cigarette users and not combustible cigarette smokers
We agree that the text would read better with the suggestion and the phrase has been changed.
L.200: add clinical studies on lung function alterations by e-cigarette use (Respirology. 2018;23(3):291–297. doi: 10.1111/resp.13180)
Reference has been added as suggested by the reviewer.
L.208-209: suggested to read: however the etiology of this acute illness is still unclear and under ongoing scientific investigation
The suggestion has been incorporated into the text.
L.210: it is not a cessation tool, since it promotes addiction therefore it is suggested to read as: In addition to previous studies, the present study shows that e-cigarettes pose a significant risk for lung health 
(relevant refference needed)
We thank the reviewer for the suggestion. Since the scope of the current manuscript is not on the controversy surrounding the use of e-cigarettes as a smoking cessation tool, we prefer to leave the current sentence as stated.
L.212 -214: it is suggested not to compare e-cigarette with combustible cigarette, but to refer instead clinical studies that show e-cigarette effects on lung function such as:Respirology. 2018;23(3):291–297. doi: 10.1111/resp.13180, Toxicol. Appl. Pharmacol. 2014; 278: 9– 15. and Tob. Prev. Cessat. 2017; 3: 1-8
We agree with the reviewer that e-cigarettes affect lung function, but there are multiple reports in the scientific literature comparing e-cigarette use to conventional cigarettes , particularly as manufacturers of e-cigarettes have marketed them as safer than conventional cigarettes, that is why we mentioned the use of e-cigarettes as a smoking cessation tool. We thank the reviewer for the suggested reference and have added them to the manuscript.
Reviewer 4 Report
This study is extremely timely, with more deaths being attributed to vaping/e-cigarettes seemingly daily. So, the strong evidence presented here, even with its admitted limitations, of a link between e-cigs and COPD should be widely-covered not only in scientific literature, but also by mainstream media.
Author Response
We want to thank reviewer 4 for the kind words and encouragement. We also agree that these findings are important and this information needs to be made available to the scientific community.
Reviewer 5 Report
Authors have rigorously interrogated from a large data set (ref. 10) of 32K US adults information related to e-cig users and non-users and, using a complex survey design, have reported the incidents of self-reported COPD in 2,727, of 3,642 e-cig users and matched controls, data reportedly being matched for active smokers and “non-smokers”. The author concluded that e-cig use was associated with a higher reported diagnosis of COPD even among a non-smoking subgroup of e-cig users. The paper reports carefully analyzed available but suboptimal data collected from a large population and puts forth the well-documented conclusion that the 10-15% of adults who use e-cig have a considerable incidence of self-reported COPD compared to a matched group on non e-cigarette users. Authors are to be congratulated for their excellent and helpful collection of references.
Comments
A major problem relates to the relative lack of distinction, in many places in the manuscript including line 23 of Abstract, of what is meant by a “non-smoker” (e.g., is this a lifetime non-smoker; what % of the “adult” e-cig smokers are past + current smokers?) Missing data included % of e-smokers who switched from active smokers to non-smokers because of having COPD and what % of the e-smokers never smoked cigarettes. This needs to be detailed in the Discussion. Abstract needs to be more specific. What % of the adult e-smokers had COPD and what % were ever smokers? What were the precise dates of the data collection? Print size of Table 1 too small to easily read and would consider either simplification or making available as a supplement. Was some of this data already in ref. 10? Figures 2 and 3 need to mention “n” on either figure or figure legend. Line 193, “self-reported diagnosis of COPD” is better. A few of the references need to be more complete. For example, 38, 40, possibly 48.
Author Response
We thank reviewer 5 for the suggestions and comments provided in order to make our manuscript better. We will have addressed the comments and suggestions as follows:
Authors have rigorously interrogated from a large data set (ref. 10) of 32K US adults information related to e-cig users and non-users and, using a complex survey design, have reported the incidents of self-reported COPD in 2,727, of 3,642 e-cig users and matched controls, data reportedly being matched for active smokers and “non-smokers”. The author concluded that e-cig use was associated with a higher reported diagnosis of COPD even among a non-smoking subgroup of e-cig users. The paper reports carefully analyzed available but suboptimal data collected from a large population and puts forth the well-documented conclusion that the 10-15% of adults who use e-cig have a considerable incidence of self-reported COPD compared to a matched group on non e-cigarette users. Authors are to be congratulated for their excellent and helpful collection of references.
Comments
A major problem relates to the relative lack of distinction, in many places in the manuscript including line 23 of Abstract, of what is meant by a “non-smoker” (e.g., is this a lifetime non-smoker; what % of the “adult” e-cig smokers are past + current smokers?)
We want to also thank reviewer 5 for the input provided. We will address his/her comments under the assumption that by e-smoker the reviewer is referring to e-cigarette users.
We apologize for the confusion, line 94 of the original submission defines nonsmoker in the following statement: “Nonsmokers were defined as those who smoked fewer than 100 cigarettes in their lifetime, a standard definition used by the Centers for Disease Control and Prevention.” We apologize for the confusion, line 145 -147, of the original submission contains the requested proportions. Among current e-cigarette users, 69.7% (95% confidence interval [CI] 68.0-71.4) were current smokers, 14.3% (95% CI 12.9 - 15.7) were former smokers and 16.0% (95% CI 14.9-17.1) were nonsmokers.
Missing data included % of e-smokers who switched from active smokers to non-smokers because of having COPD and what % of the e-smokers never smoked cigarettes. This needs to be detailed in the Discussion.
We thank the reviewer for this suggestion, unfortunately, PATH survey does not have information on the number of participants that stopped e-cigarette and/or cigarette use or switched from cigarettes to e-cigarettes and vice versa after a diagnosis of COPD. We have included this in the limitation.
On line 147 of original submission we reported the percentage of e-cigarette users that were nonsmokers. Among e-cigarette users, 16.0 (95% CI 14.9-17.1) were nonsmokers.
Abstract needs to be more specific. What % of the adult e-smokers had COPD and what % were ever smokers?
We apologize for the confusion. The percentage of current e-cigarette smokers that reported a diagnosis of COPD was 11.1% (95% CI 9.7-12.5) and this was reported on Line 173 of the original submission.
We apologize for the confusion. As we stated above, nonsmokers were defined as those who smoked fewer than 100 cigarettes in their lifetime, a standard definition used by the Centers for Disease Control and Prevention [36], we included the percentage of e-cigarette users that were nonsmokers on line 147 of original submission. Among e-cigarette users, 16.0 (95% CI 14.9-17.1) were nonsmokers.
We could not incorporate these percentages into the abstract due to word count limitation.
What were the precise dates of the data collection?
We apologize for the confusion, the interval dates of data collection are presented in line 69 of the original submission. The data was collected from September 12, 2013 to December 15, 2014
Print size of Table 1 too small to easily read and would consider either simplification or making available as a supplement. Was some of this data already in ref. 10?
We apologize for this and have increased the font of the table. Table 1 refers to our propensity match cohort, which is not part of the report by Coleman et al therefore we included this in the manuscript for clarity.
Figures 2 and 3 need to mention “n” on either figure or figure legend.
We thank the reviewer for this suggestion, but we believe that adding the number of participants to Figure 2 will make this unreadable. There is no Figure 3.
Line 193, “self-reported diagnosis of COPD” is better. A few of the references need to be more complete. For example, 38, 40, possibly 48.
We have modified this sentence. The references have been edited and entered using Endnote® and reviewed for accuracy.